# ATTNINPUT: REVOLUTIONIZING PINYIN INPUT WITH CONTEXT-AWARE RWKV LANGUAGE MODELS

## ABSTRACT

The Pinyin Input Method Engine (IME) is widely used for inputting Chinese characters, but effectively integrating it with powerful large language models (LLMs) remains a challenge due to issues such as semantic discontinuity and inefficient training. This paper presents AttnInput, a novel approach that leverages the strengths of the RWKV language model, specifically its linear computational complexity and "infinite" context length, to enhance Pinyin IME. Our method integrates Pinyin information directly into the internal state of RWKV through a lightweight side network, effectively addressing the semantic discontinuity issue faced by previous LLM-based IMEs. Furthermore, AttnInput utilizes a pre-training strategy, significantly reducing training data and computational costs compared to previous methods. Experimental results demonstrate that AttnInput achieves state-of-the-art performance on abbreviated Pinyin input, especially as the Pinyin sequence length increases. This efficient design allows us to scale up to larger models and incorporate longer contexts, further improving accuracy and user experience.

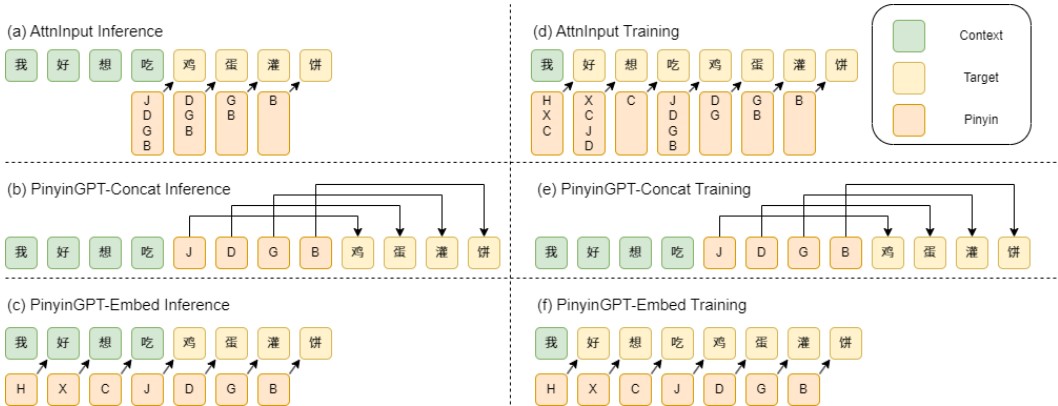

Figure 1: Illustration of the inference and training process of pinyin IMEs. The abbreviated pinyin of the Chinese characters "我好想吃鸡蛋灌饼"(I really want to eat an egg pancake) shown in the picture is "W H X C J D G B". See Appendix B for detailed information.

## 1 INTRODUCTION

Pinyin Input Method Engine (IME) allows users to input Chinese characters using a standard keyboard. Pinyin is the official romanization system for Chinese, which represents the pronunciation of Chinese characters using the Latin alphabet.

The advent of GPT models has spurred research into applying large language models to input method engines. As illustrated in Figure 1(b), most of the previous research that achieve state-of-the-art performance like PinyinGPT-Concat (Tan et al., 2022) and GeneInput (Ding et al., 2023) simply concatenate the context and the pinyin sequence to form the prompt for the language model. However, inserting pinyin sequences disrupts the semantic flow between the prompt and target text, and

poses challenges for effectively leveraging pretrained large language models, as their training objective primarily focuses on predicting the next token. Furthermore, these models are trained in an SFT manner, indicating that only a small number of pinyin information in each training sample is learned, leading to a need for extensive training resources and difficulty in increasing context length. Our work confirms that concat-based method disrupts semantic consistency and leads to inefficient training. As illustrated in Figure 1(c), pinyinGPT-embed (Tan et al., 2022) demonstrates superior training efficiency, however, its performance remains suboptimal due to its inability to fully utilize the pinyin information in the input during inference.

We explored the direct use of pinyin-constrained beam search outputs from large language models as candidate word lists, resulting in substantial performance improvement. Nevertheless, this method abandons pinyin information, which leads to a higher probability of prematurely pruning the correct answer during the initial stages of beam search, particularly when the target's prefix tokens are infrequent. This presents opportunities for further improvement.

Therefore, we propose a novel approach named AttnInput to leverage large language models for input method engine. It addresses the semantic discontinuity of previous methods by integrating Pinyin information directly into the RWKV's internal state through a lightweight side network. This side network uses ladder side-tuning, attaching to the main model without requiring backpropagation through it, thus saving computational resources. The model is pre-trained, unlike many previous approaches which use fine-tuning, leading to more efficient use of training data and lower computational cost. During inference, the model receives both the context and a sequence of abbreviated Pinyin, processing them together to predict the corresponding Chinese characters. The use of RWKV allows for efficient handling of long contexts and Pinyin sequences. Pinyin-constrained training and beam search are employed to further improve accuracy by restricting predictions to characters matching the given Pinyin. AttnInput offers the following advantages:

- To the best of our knowledge, it achieves state-of-the-art performance on abbreviated pinyin.

- In the training stage, it requires significantly less computational resources and training data compared to previous work.

- It is based on RWKV6(Peng et al., 2024), a linear attention large language model, which is more suitable for input method engine due to its "infinite" context length[1] and efficiency in inference.

## 2 TASK

The input of pinyin input method includes a sequence of Chinese characters $W = \{w_1, ..., w_n\}$ representing the context and a sequence of abbreviated pinyin $P = \{p_1, ..., p_m\}$. Each abbreviated pinyin is a single English letter, ranging from a to z. The output is a sequence of Chinese characters $O = \{w_{n+1}, ..., w_{n+m}\}$. The output sequence follows the input sequence semantically, and the pronunciation corresponds to the abbreviated pinyin.

## 3 MODELS

In this section, we first introduce standard RWKV6 large language model. The vanilla RWKV6 model exhibits competitive performance compared to existing state-of-the-art models in IME tasks, even when ignoring pinyin information during inference. Afterward, we will introduce the new model named AttnInput, which can leverage enriched pinyin information during inference while maintaining efficient training and inference performance.

---

[1]The authors of RWKV6 claim that RWKV6 has "infinite" context length on https://rwkv.com/ due to the observed continuous decrease in loss as the context length extends beyond the context length used during training. However, this does not necessarily imply that RWKV6 outperforms Transformer-based models in long-text understanding or retrieval tasks.

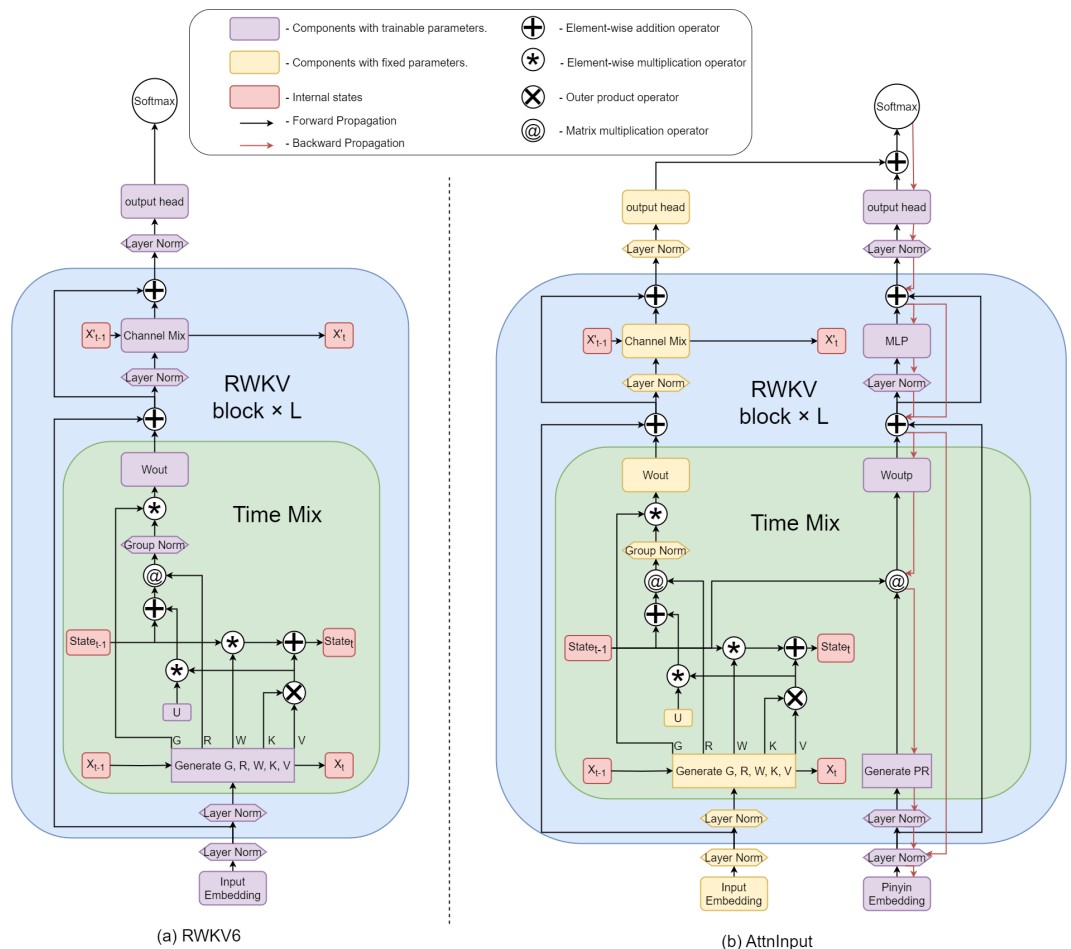

Figure 2: Architecture of the RWKV6 and proposed model, AttnInput.

## 3.1 RWKV6

As illustrated in Figure 2(a), we choose RWKV6 as the backbone, which is a RNN with performance comparable to Transformer-based LMs. The RWKV attention *aka* Time Mix can be written in a recurrent manner:

$$\boldsymbol{X} = \boldsymbol{v}^T \otimes \boldsymbol{k} \tag{1}$$

$$\boldsymbol{S}' = \boldsymbol{S} \otimes \mathrm{diag}(\boldsymbol{w}) + \boldsymbol{X} \tag{2}$$

$$\boldsymbol{y} = (\boldsymbol{X} \otimes \mathrm{diag}(\boldsymbol{u}) + \boldsymbol{S}) \otimes \boldsymbol{r} \tag{3}$$

In which, $\otimes$ is matrix multiplication operator, $\boldsymbol{S}$ is the internal state that similar to the KVCache in the Transformer, but has a constant size, $\boldsymbol{r}$ controls forgetting, $\boldsymbol{w}$ controls attention, $\boldsymbol{k}$ and $\boldsymbol{v}$ store and retrieve information, $\boldsymbol{u}$ is content-dependent bias.

## 3.2 ATTNINPUT

As illustrated in Figure 2(b), we introduce the new model named AttnInput. We use the RWKV6 model as the backbone model and attach a relatively small side network to the backbone model to extract the pinyin feature and integrate it with information from the context.

We integrate pinyin feature with context information by mapping the former to a fixed-size vector through a linear layer and multiplying it with the internal state of the RWKV6 model. The formula is as follows:

$$py = S' \otimes pr \qquad (4)$$

In which, $py$ is the pinyin-state mixed information, $pr$ is a vector generated from the pinyin information, and $S'$ is the internal state.

## 3.3 LADDER SIDE-TUNING

As illustrated in Figure 2(b), we employ ladder side-tuning (Sung et al., 2022) to attach side networks for mixing pinyin and context information. This approach avoids backpropagating updated parameters through the backbone network.

Due to the significantly fewer parameters in the side network compared to the backbone network, it can save a large amount of computation and memory usage for storing activation values, gradients and optimizer states. See Appendix A for the detailed cost analysis.

## 3.4 ENCODING PINYIN SEQUENCE

As illustrated in Figure 1(a), for a certain position $i$ in the pinyin sequence, we select this position and subsequent pinyin $P_i = \{p_i, ..., p_m\}$ as the pinyin information input to the model at this position. Therefore, there is no information interaction between the pinyin information at different positions in the input. The output $O_i$ at each position $i$ is only related to the text context $W_i = \{w_1, ..., w_{n+i-1}\}$ and the pinyin information $P_i$. This ensures the efficiency of training, as each character's pinyin information is trained, while also maintaining consistency in the data input during both training and inference.

To encode the pinyin sequence, we employ a concatenation operation to combine all pinyin embedding vectors into a unified representation. We pad pinyin sequences with zeros to a fixed length, which is 16 in our experiments. Sequences exceeding this length are truncated. We tokenize pinyin sequence by mapping each letter to its position in the alphabet.

## 3.5 EFFICIENT TRAINING

As illustrated in Figure 1(e), the AttnInput model is trained in a pre-training manner, which is similar to the one used in the large language models. The pinyin sequences at each position are independent, with no information interaction between them, to ensure consistency during training and inference. This method potentially enables the model to leverage pinyin information from a greater proportion of tokens within the training data.

However, for previous concat-based models like PinyinGPT-Concat and GeneInput, the design that connects pinyin to the context makes it necessary to train them using the SFT method, as shown in Figure 1(f). Assuming that the length of the context in the training data is $n$ and the length of the pinyin is $m$, with $n$ being much larger than $m$, only the pinyin information of $m$ tokens will be learned. This suggests that AttnInput potentially exhibits a $\frac{n}{m}$ times improvement in training data utilization compared to prior approaches.

## 3.6 PINYIN-CONSTRAINED TRAINING AND INFERENCE

The model is trained using the Pinyin-Constrained Training (Tan et al., 2022) method. The probability distribution for the next Chinese characters is calculated solely over Chinese characters that perfectly match the pinyin. The formula is as follows:

$$P(w_i|\boldsymbol{w}_{<i}, \boldsymbol{p}_i) = \frac{\exp(g(w_i|\boldsymbol{w}_{<i}, \boldsymbol{p}_i))}{\sum_{w \in V_{\boldsymbol{p}_{i,0}}} \exp(g(w_i|\boldsymbol{w}_{<i}, \boldsymbol{p}_i))} \qquad (5)$$

where $g$ is the output of the model, $\boldsymbol{p}_i$ is the pinyin sequence at position $i$, $V_{\boldsymbol{p}_{i,0}}$ is the set of all possible Chinese characters that match the abbreviated pinyin sequence $p_i$, and $\boldsymbol{w}_{<i}$ is the context up to position $i$.

Since abbreviated pinyin can correspond to multiple Chinese characters, for those models mentioned in this paper including AttnInput, PinyinGPT-Concat, vanilla RWKV6, and RWKV6-concat-lora,

we use beam search to generate possible character sequences. Each token is generated in a auto-regressive manner, and only those Chinese characters that perfectly correspond to the pinyin are considered, in order to improve accuracy. The detailed formula is presented in 5.

## 4 EXPERIMENT

### 4.1 SETTINGS

#### 4.1.1 DATASET

We use SkyPile-150B (Wei et al., 2023) to generate training and evaluation dataset, which is a large-scale and comprehensive Chinese dataset including 150 billion tokens and 620 gigabytes of text data. SkyPile-150B is not included in the training datasets of the RWKV6 models. The corresponding abbreviated pinyin sequences are automatically generated using the public Python library, pypinyin[2].

The evaluation data is derived from SkyPile-150B, with pinyin lengths ranging from 1 to 16 and context lengths of 64 ,512 and 1536. Each evaluation set contains 500 context-pinyin pairs, which are strictly separated from the training data.

#### 4.1.2 TRAINING

We use RWKV6-1.6B, a pretrained RWKV6 model with 1.6B parameters, as the backbone model, which is fixed during training. AttnInput have a side network with 500M trainable parameters. The loss function is cross-entropy loss. The max learning rate is 3e-4. The learning rate is decayed by cosine annealing with a warmup period of 300 steps. The optimizer is AdamW with a weight decay of 0.01. The batch size is 8. The context length is 1024. The length of pinyin sequence at each position is randomly selected from $[0, 16]$. The model is trained for 40K steps on a single RTX 4090D GPU.

To ensure a fair comparison with previous concat-based methods, we also trained a concat-based model with RWKV6-1.6B, labeled as RWKV6-concat-lora. This model was fine-tuned with LoRA (Hu et al., 2021) and includes 500M trainable parameters. The training data is the same as the AttnInput model.

#### 4.1.3 EVALUATION METRIC

We use the precision at top-K as the evaluation metric, which measures if the ground-truth Chinese character sequence is among the top-K predicted sequences. K is set to 1, 5, 10, and 15.

### 4.2 RESULTS

In this section, we will present the results of the proposed models for abbreviated pinyin on the SkyPile-150B dataset. We compare AttnInput with vanilla RWKV6, PinyinGPT-Concat and RWKV6-concat-lora. GeneInput is not included as its source code or datasets are not publicly released and it do not show better performance than PinyinGPT-Concat on abbreviated pinyin. All outputs are generated by Pinyin-Constrained beam search, with a beam size of 16. When testing pinyinGPT-concat, we used a context window of size 128, as it was trained on text that does not exceed 128 tokens. The context lengths of 64, 512, and 1536 represent cases of short text, long text, and text exceeding the context window, respectively.

Figure 3 demonstrates that the proposed AttnInput model consistently outperforms vanilla RWKV6, PinyinGPT-Concat and RWKV6-concat-lora across most pinyin and context lengths. Several key findings emerge from the results.

- We can see that when the length of the pinyin sequence increases, the performance advantage of AttnInput over vanilla RWKV6 becomes increasingly significant, as the proposed model can leverage more information from the pinyin sequence to generate more accurate Chinese characters.

---

[2]https://pypi.org/project/pypinyin

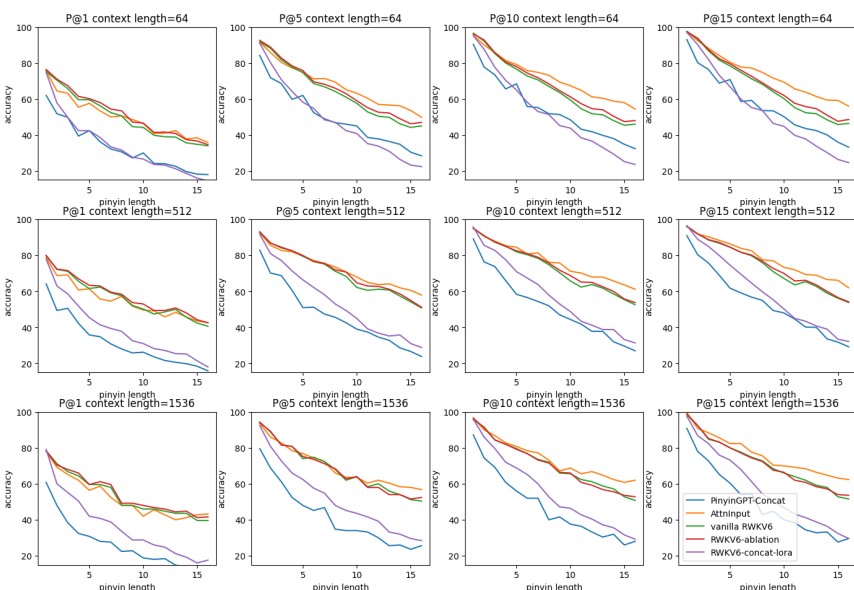

Figure 3: Evaluation results of the proposed model. The x-axis represents the length of the pinyin sequence, and the y-axis represents the Top-K accuracy of the model, with K=1, 5, 10, and 15. The context lengths for the three rows are 64, 512, and 1536, respectively. Detailed numeric results are shown in 3.

- All models exhibit decreasing accuracy with increasing pinyin sequence length. This is attributable to the exponential growth in possible character sequences matching a given abbreviated pinyin sequence, increasing ambiguity.

- Leveraging longer contexts significantly benefits both AttnInput and the vanilla RWKV6, likely due to the richer information available in such contexts, including names and locations challenging to infer from pinyin alone. However, PinyinGPT-Concat, trained on contexts shorter than 128 tokens, struggles to exploit this additional information effectively.

- AttnInput exhibits strong length extrapolation capabilities, maintaining superior performance compared to other models even when the context length exceeds the context window.

- The observed inferior performance of RWKV6-concat-lora relative to vanilla RWKV6 provides compelling evidence in support of our proposition that concat-based method disrupts semantic consistency and leads to inefficient training.

## 4.3 ANALYSIS AND DISCUSSION

We noticed that AttnInput performs slightly worse than vanilla RWKV6 in Top-1 accuracy. This phenomenon is also observed in previous works (Tan et al., 2022). Our hypothesis is that the training procedure led to a slight degradation in the original model's performance. We analyzed instances where the vanilla RWKV6 model provided the correct answer, while AttnInput failed to prioritize the target. Our investigation revealed that in these specific instances, the abbreviated pinyin corresponded to numerous contextually appropriate Chinese character sequences, causing AttnInput to encounter difficulties in accurately ranking them based on probability. This observation supports our initial hypothesis.

The performance gains observed in other metrics are hypothesized to be a consequence of AttnInput boosting the scores of the initial target tokens based on pinyin information. This mechanism effectively prevents the early elimination of potential target sequences during beam search, especially when the initial tokens are relatively rare.

Table 1: Case study on abbreviated pinyin

| Case | Predictions | | |
|---|---|---|---|
| | **PinyinGPT -Concat** | **vanilla RWKV6** | **AttnInput** |
| **Context:** 1998年 **Target:** 汉城举办的第 二十四届奥运会 **Pinyin:** HCJBDD ESSJAYH **Translation:** The 24th Olympic Games held in Seoul in 1998 | 汉城举办的 第二十三届 奥运会 (The 23rd Olympic Games held in Seoul) | 后重建并得 到二十四届 奥运会 (After reconstruction and getting the 24th Olympic Games) | 汉城举办的 第二十四届 奥运会 (The 24th Olympic Games held in Seoul) |
| **Context:** 首先，问问目前A股 市场的大多数投资者：你选择 购买股票还是基金？阅读 下面的新闻可能会有帮助。 **Target:** 开源证券最近 发布了一份报告 **Pinyin:** KYZQZJFBLYFBG **Translation:** Firstly, ask most investors in the current A-share market: Do you choose to buy stocks or funds? Reading the following news may be helpful. KAIYUAN Securities recently released a report | 开源证券 最近发布了 一份报告 (KAIYUAN Securities recently released a report) | 可以在其中 就发布了 一份报告 (A report can be published within it) | 开源证券 最近发布了 一份报告 (KAIYUAN Securities recently released a report) |
| **Context:** 磁性测厚法:适用导磁 **Target:** 材料上的非导磁层厚度 **Pinyin:** CLSDFDCCHD **Translation:** Magnetic thickness measurement method: applicable to the thickness of non-magnetic layers on magnetic materials | 材料深度放 大尺寸厚度 (Material depth, enlarged size, thickness) | 材料上的非 导磁层厚度 (Thickness of non-magnetic layer on material) | 材料上的非 导磁层厚度 (Thickness of non-magnetic layer on material) |

## 4.4 ABLATION STUDY

This section describes an ablation study designed to confirm whether the model learns the inherent relationship between pinyin and text, as opposed to simply improving its general Chinese language modeling ability. We use the same model configuration, training setup, and dataset as before, but replace the pinyin sequences with blank ones to ensure the model does not learn from pinyin information.

As shown in Figure 3, although this model performs slightly better than the original, it still significantly underperforms compared to AttnInput, especially for longer pinyin sequences, indicating that AttnInput indeed learns and utilizes the information from the pinyin.

## 4.5 CASE STUDY

We list three cases in Table 1 to compare outputs produced by PinyinGPT-Concat, vanilla RWKV6, and AttnInput. In case 1 and 2, the vanilla RWKV6 fails to generate the correct answer due to the presence of uncommon characters at the beginning, whereas PinyinGPT-Concat and AttnInput succeed by utilizing pinyin information. In case 1 and 3, PinyinGPT-Concat fails as it lacks the necessary common-sense knowledge. Notably, in all cases, AttnInput consistently produces the correct output.

## 4.6 LATENCY ANALYSIS

To apply the proposed model to real-world scenarios, we need to analyze its latency. Since the context only expands at the end during the input process, we cache the internal state to avoid repeated prefill operations. Therefore, the latency is equal to the time it takes to generate one token multiplied by the length of the pinyin sequence. We tested the time it takes to generate a token under different beam size settings on a single RTX 4090D GPU, the results are summarized in Table 2.

Table 2: The time it takes to generate one token under different beam size settings

| beam size | time (ms) |
| --- | --- |
| 4 | 19.06 |
| 8 | 19.00 |
| 16 | 19.53 |
| 24 | 24.06 |
| 32 | 29.08 |

As we can see, with a beam size of 16, the latency is approximately 20ms. Assuming the user inputs a pinyin sequence of length 4, the latency would be 80ms, which is practical for real-world scenarios. The latency can be further optimized by using a smaller model or a faster GPU.

## 5 RELATED WORKS

### 5.1 CLASSICAL PINYIN IMES

Pinyin Input Method Engines (IMEs) have been extensively studied for decades, with a focus on improving accuracy and efficiency. Early methods relied heavily on statistical language models, such as N-gram models (Chen & Lee, 2000), statistical machine translation (Yang et al., 2012) and Conditional Random Fields (Xia & Cheung, 2016). These approaches often struggled with data sparsity and lacked the ability to capture long-range dependencies in language.

### 5.2 NEURAL PINYIN IMES

Recent years have witnessed the successful application of neural networks to Pinyin IMEs. Long Short-Term Memory (LSTM) networks (Zhang et al., 2019; Huang & Zhao, 2018) and attention-based neural networks (Huang et al., 2018) have achieved promising results by modeling sequential data effectively. However, these models face limitations in capturing long-term dependencies and parallelization during training.

### 5.3 LARGE LANGUAGE MODELS FOR PINYIN IMES

The emergence of large language models (LLMs) like GPT has opened up new possibilities for Pinyin IMEs. Recent work has explored the use of LLMs for generating candidate characters based on Pinyin input (Tan et al., 2022; Ding et al., 2023). However, directly applying LLMs to Pinyin IMEs presents challenges, including semantic discontinuity caused by inserting Pinyin sequences and the need for large amounts of training data and computational resources. Our work differs from previous works in that we are the first one to fully leverage the power of large language models and train the models to learn pinyin-context relationships efficiently in a pre-training manner, achieving state-of-the-art performance with minimal training data and computational resources.

## 6 CONCLUSION

This paper introduces AttnInput, a novel approach for Pinyin IME that effectively integrates Pinyin information with a large language model, RWKV, for accurate and efficient Chinese character prediction. By addressing semantic discontinuity and reducing computational overhead, AttnInput achieves state-of-the-art performance on abbreviated Pinyin input. Moreover, the efficient design of

AttnInput allows for scaling up to larger models and incorporating longer contexts, paving the way for even more accurate and context-aware Pinyin input methods. This work signifies a significant step towards more powerful and efficient integration of LLMs within IMEs, ultimately improving user experience.

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

## A  COMPUTING COST

Throught out this section, we denote by $N$ the total number of parameters in the backbone RWKV6 model, $M$ the total number of parameters in the side network, $L$ the number of layers, $h$ the number of heads and $d = 64$ the dimension of each head. All models are trained with $h = 32$, $L = 24$, $N = 1.6B$ and $M = 500M$.

The inference FLOPs for each tokenis approximated as follows:

$$\#(\text{InferFLOPs}) = 2(N + M) + 9d^2hL \tag{6}$$

since each matrix requires one multiplication and one addition operation and the rwkv attention requires $9d^2h$ operations(see 1 2 3 4).

The training FLOPs for each token is approximated as inference FLOPs plus four times the total number of trainable parameters plus the FLOPs for backpropagating in rwkv attention:

$$\#(\text{TrainFLOPs})_{\text{L}} = 2N + 6M + 14d^2hL \tag{7}$$

In Full fine-tuning, all parameters are updated, so the training FLOPs for each token is approximated as follows:

$$\#(\text{TrainFLOPs})_{\text{F}} = 6N + 6M + 21d^2hL \tag{8}$$

$$1 - \frac{\#(\text{TrainFLOPs})_{\text{L}}}{\#(\text{TrainFLOPs})_{\text{F}}} = 0.507 \tag{9}$$

That is, ladder side-tuning saves 50.7% FLOPs in training compared to full fine-tuning.

## B  A BRIEF INTRODUCTION TO HANYU PINYIN AND ITS ROLE IN CHINESE TEXT INPUT

Hanyu Pinyin, or Pinyin, is the standard romanization system for Standard Mandarin Chinese. It employs the Latin alphabet to represent the sounds of Mandarin, aiding in pronunciation and language learning. Importantly, Pinyin is not a replacement for Chinese characters, which are the core written units conveying meaning in the language.

The relationship between Pinyin and Chinese characters can be summarized as:

- Characters as Semantic Units: Chinese characters are primarily logographic, with each character representing a morpheme or word and carrying meaning.

- Pinyin as Phonetic Representation: Pinyin indicates the pronunciation of characters but does not convey meaning directly.

- Homophony and Context: A single Pinyin spelling can correspond to multiple characters with different meanings due to homophones (same pronunciation, different meanings). Context is crucial for disambiguation. For example, the abbreviated pinyin "JDGB" in Figure 1 can match multiple Chinese phrases, such as "鸡蛋灌饼" (egg pancake) and "见到过吧" (have you seen it before).

- Tones: Pinyin uses diacritical marks to denote the four main tones in Mandarin, which are essential for distinguishing meaning.

The advent of computers and mobile devices has made Pinyin indispensable for Chinese text input. Pinyin input methods allow users to type Pinyin on a standard keyboard and then select the corresponding Chinese characters from a list of suggestions. This technology significantly bridges the gap between the phonetic representation of Pinyin and the character-based writing system.

## C  EXPERIMENT RESULTS

Table 3: The numeric table of 3. To keep the table concise, only the average scores across consecutive sets of four lengths are shown.

| Context Length | Pinyin Length | Evaluation Metric | PinyinGPT -Concat | Attn Input | Vanilla RWKV6 | RWKV6- ablation | RWKV6 -concat -lora |
|---|---|---|---|---|---|---|---|
| 64 | 1-4 | P@1 | 50.8 | 64.8 | 67.8 | **69.0** | 56.2 |
| | | P@5 | 71.1 | 83.7 | 84.9 | **85.7** | 76.5 |
| | | P@10 | 76.8 | 88.2 | 88.5 | **88.8** | 82.7 |
| | | P@15 | 79.6 | **90.5** | 89.8 | 90.2 | 85.6 |
| | 5-8 | P@1 | 35.4 | 52.9 | 54.7 | **56.5** | 36.5 |
| | | P@5 | 52.5 | **71.8** | 68.5 | 69.9 | 52.1 |
| | | P@10 | 57.8 | **75.8** | 71.8 | 73.2 | 56.9 |
| | | P@15 | 60.6 | **77.5** | 73.1 | 74.3 | 58.9 |
| | 9-12 | P@1 | 26.4 | **44.2** | 41.9 | 44.2 | 25.3 |
| | | P@5 | 41.8 | **61.5** | 55.4 | 57.4 | 38.0 |
| | | P@10 | 46.2 | **65.7** | 57.3 | 59.4 | 41.0 |
| | | P@15 | 48.2 | **67.6** | 58.0 | 60.2 | 42.2 |
| | 13-16 | P@1 | 19.6 | **38.6** | 35.8 | 37.3 | 17.6 |
| | | P@5 | 32.5 | **54.0** | 46.3 | 48.6 | 25.8 |
| | | P@10 | 36.2 | **57.9** | 47.6 | 50.0 | 27.8 |
| | | P@15 | 37.9 | **59.0** | 48.1 | 50.4 | 28.8 |
| 512 | 1-4 | P@1 | 51.6 | 69.4 | 72.2 | **72.8** | 62.8 |
| | | P@5 | 70.7 | 85.8 | 86.7 | **86.8** | 80.3 |
| | | P@10 | 76.4 | **89.8** | 89.6 | 89.6 | 85.4 |
| | | P@15 | 79.0 | **91.8** | 90.9 | 91.0 | 87.7 |
| | 5-8 | P@1 | 32.3 | 57.2 | 60.2 | **61.1** | 41.0 |
| | | P@5 | 48.8 | **76.5** | 75.6 | 76.0 | 60.0 |
| | | P@10 | 55.4 | **80.8** | 78.9 | 79.6 | 65.0 |
| | | P@15 | 58.2 | **82.7** | 80.5 | 80.8 | 67.1 |
| | 9-12 | P@1 | 24.3 | 49.1 | 49.6 | **51.4** | 29.8 |
| | | P@5 | 38.4 | **66.9** | 63.1 | 65.3 | 42.6 |
| | | P@10 | 42.8 | **71.3** | 65.6 | 67.7 | 46.6 |
| | | P@15 | 45.5 | **73.0** | 66.8 | 68.7 | 48.5 |
| | 13-16 | P@1 | 18.7 | 45.1 | 44.6 | **46.4** | 22.5 |
| | | P@5 | 27.9 | **61.2** | 55.8 | 56.5 | 32.7 |
| | | P@10 | 31.6 | **64.7** | 57.1 | 58.0 | 35.5 |
| | | P@15 | 33.6 | **66.0** | 58.0 | 58.6 | 36.3 |
| 1536 | 1-4 | P@1 | 44.9 | 68.6 | 70.2 | **70.7** | 61.2 |
| | | P@5 | 65.5 | 85.2 | **86.4** | 86.4 | 78.0 |
| | | P@10 | 72.9 | **89.1** | 88.4 | 88.5 | 83.4 |
| | | P@15 | 76.7 | **91.1** | 89.7 | 89.9 | 85.7 |
| | 5-8 | P@1 | 27.2 | 53.9 | 56.3 | **57.4** | 38.8 |
| | | P@5 | 43.7 | **72.4** | 72.3 | 72.1 | 55.7 |
| | | P@10 | 50.0 | **77.4** | 75.5 | 75.2 | 61.7 |
| | | P@15 | 52.9 | **79.5** | 76.3 | 76.0 | 64.2 |
| | 9-12 | P@1 | 19.5 | 44.6 | 46.2 | **47.5** | 27.1 |
| | | P@5 | 32.8 | **62.4** | 61.1 | 60.7 | 42.4 |
| | | P@10 | 37.2 | **67.1** | 63.8 | 63.1 | 44.2 |
| | | P@15 | 39.4 | **69.5** | 65.1 | 64.2 | 45.4 |
| | 13-16 | P@1 | 13.9 | 41.8 | 41.6 | **43.0** | 18.5 |
| | | P@5 | 25.2 | **58.4** | 52.9 | 53.0 | 30.8 |
| | | P@10 | 29.1 | **62.5** | 54.9 | 54.7 | 33.4 |
| | | P@15 | 30.8 | **64.2** | 55.4 | 55.8 | 34.3 |

