# OpenReview forum: "AttnInput: Revolutionizing Pinyin Input with Context-Aware RWKV Language Models"
_ICLR.cc/2025/Conference — ICLR 2025 Conference Withdrawn Submission_

### Official Review · Reviewer_gCM1 · 2024-10-16

**Soundness:** 2
**Presentation:** 1
**Contribution:** 2
**Rating:** 5
**Confidence:** 3

**Summary:**

This paper introduces AttnInput, a novel approach integrating Pinyin information with RWKV-6. The method combines contextual and Pinyin information with a side network, enabling more efficient training than traditional techniques. Experimental results demonstrate that AttnInput achieves state-of-the-art performance on abbreviated Pinyin input.

**Strengths:**

1. The paper introduces a method that employs RWKV-6 for Pinyin IMEs. RWKV, an RNN-based model, is particularly well-suited for this task.
2. The design of the side model efficiently integrates Pinyin information into the backbone model, and ladder side-tuning enables efficient training.
3. The proposed method achieves state-of-the-art performance in this task.

**Weaknesses:**

1. The authors demonstrate that their method is both efficient and effective for training. However, they do not provide experiments to substantiate its superiority over other Pinyin integration methods.
2. The comparison of your model with other approaches is not fair due to the inherent strengths of RWKV-6. A more appropriate comparison would involve other methods that integrate contextual and Pinyin information under the same conditions, such as comparable models.
3. The paper's presentation is unclear and lacks structure. The authors do not clearly explain the design of AttnInput, particularly the side network, nor do they describe how vanilla RWKV-6 is employed. Furthermore, the Introduction section is too brief and fails to provide an overview of the method.
4. The tables could be improved aesthetically: Table 1 exceeds the linewidth, and Table 2 would benefit from being presented in a three-line table format.

**Questions:**

1. As discussed in Weaknesses 1 and 2, what are the experimental results of other integration methods with RWKV-6?
2. The P@1 performance of the method is not better than the vanilla RWKV-6. Does this imply that incorporating Pinyin information merely imposes a constraint on the decoding process?
3. The authors claim that RWKV-6 was chosen for its infinite context window. However, RNN-based models also suffer performance degradation beyond their context window. Could the authors provide details on the length extrapolation capacity (i.e., performance beyond the context window) of their methods?

---

> ### Author Response · Authors · 2024-11-21
>
> We thank you for the feedback and address all remaining concerns below. For further details, please refer to the newly uploaded file, where the modifications are highlighted in blue font. Thank you again for your valuable comments.
>
> ## W1 & W2 & Q1
>
> This was indeed an oversight on our part. We appreciate you bringing it to our attention.
>
> To ensure a fair comparison with previous concat-based methods, we trained a concat-based model with RWKV6-1.6B, labeled as RWKV6-concat-lora. This model was fine-tuned with LoRA  and includes 500M trainable parameters. The training data is the same as the AttnInput model. We tested this model, and its performance was disappointing.
>
> The observed inferior performance of RWKV6-concat-lora relative to vanilla RWKV6 provides compelling evidence in support of our proposition that concat-based method disrupts semantic consistency and leads to inefficient training.
>
> ## Q2
>
> First, It's impossible that AttnInput only introduces constraints at the decoding stage, as constraints are inherently and explicitly applied during decoding.(see 3.6 PINYIN-CONSTRAINED TRAINING AND INFERENCE for detailed information) If AttnInput's function within decoding is *solely* to impose these constraints, it would have no impact on the output.
>
> Second, we noticed that AttnInput performs slightly worse than vanilla RWKV6 in Top-1 accuracy. This phenomenon is also observed in previous works [1]. Our hypothesis is that the training procedure led to a slight degradation in the original model’s performance. We analyzed instances where the vanilla RWKV6 model provided the correct answer, while AttnInput failed to prioritize the target. Our investigation revealed that in these specific instances, the abbreviated pinyin corresponded to numerous contextually appropriate Chinese character sequences, causing AttnInput to encounter difficulties in accurately ranking them based on probability. This observation supports our initial hypothesis.
>
> ## Q3
>
> First, we admit that this was our oversight. The authors of RWKV6 claim that RWKV6 has ”infinite” context length on https://rwkv.com/ due to the observed continuous decrease in loss as the context length extends beyond the context length used during training. However, this does not necessarily imply that RWKV6 outperforms Transformer-based models in long-text understanding or retrieval tasks.
>
> Second, we conducted experiments on text exceeding the context length used during training, and the results demonstrate that AttnInput possesses strong length extrapolation capacity. (see 4.2 RESULTS)
>
> ## W3
>
> We apologize for any confusion caused by our unclear writing. We have substantially rewritten the entire paper.
>
> ## W4
>
> Thank you! We have improved the layout of both tables according to your suggestions.
>
> References
>
> [1] [https://aclanthology.org/2022.acl-long.133.pdf](https://aclanthology.org/2022.acl-long.133.pdf)

---

> > ### Comment · Reviewer_gCM1 · 2024-11-24
> > **Raising my score.**
> >
> > Thank you for your response! The additional experiments have addressed my questions. I will increase my rating to 5.

---

### Official Review · Reviewer_ZXk9 · 2024-10-27

**Soundness:** 3
**Presentation:** 2
**Contribution:** 3
**Rating:** 6
**Confidence:** 3

**Summary:**

This paper presents a modification of RWKV architecture to support pinyin input, by integrating the pinyin information in the internal state of RWKV through a lightweight network. The proposed method has minimal computational overhead, and shows better performance than other baselines especially for longer pinyin lengths.

**Strengths:**

- The paper addresses an important problem of adding support for pinyin method input in language models.
- The proposed method seems to perform better than other baselines, particularly on longer context lengths with P@10 and P@15 eval.
- The writing is clear overall. A few improvements (highlighted below) would further improve the clarity of the paper.

**Weaknesses:**

- The vanilla RWKV seems to perform better than the proposed AttnInput for the P@1 setting (up to 9 pinyin length) and the P@5 setting (up to 6 pinyin length).
- AttnInput is evaluated on one dataset. It should be evaluated on additional pinyin input datasets to establish the generalizability of the proposed approach.
- A comparison of the performance of AttnInput with the regular GPT baseline would be useful (similar to how it is done in the PinyinGPT paper [1]).
- Were multiple runs done (with different seeds) for the main results presented in Figure 3? If so, the error bars should be included as well.
- Minor issues:
    - Line 49: achieving -> achieve
    - Line 95 - 97: v, k and r should be defined.
    - Line 276, 286, 296: Traget -> Target
    - Line 53: , poses challenges -> and poses challenges
    - Line 80-83: This paragraph should be moved to a more relevant position.


References:

[1] https://aclanthology.org/2022.acl-long.133.pdf

**Questions:**

- While Figure 3 helps in comparing the performance trend, can the accuracy numbers be presented in a Table form as well? It would assist in quantitatively understanding the differences in performance.
- What are the potential reasons for the better performance of vanilla RWKV over the proposed approach in the P@1 setting (up to 9 pinyin length) and P@5 setting (up to 6 pinyin length)? How will this affect the practical usage of the proposed model for pinyin input?

---

> ### Author Response · Authors · 2024-11-21
>
> We thank you for the feedback and address all remaining concerns below. For further details, please refer to the newly uploaded file, where the modifications are highlighted in blue font. Thank you again for your valuable comments.
>
> ## Q1
>
> > While Figure 3 helps in comparing the performance trend, can the accuracy numbers be presented in a Table form as well? It would assist in quantitatively understanding the differences in performance.
>
> Thank you for your suggestion! The detailed numeric table is in Appendix C now.
>
> ## W1 & Q2
>
> > The vanilla RWKV seems to perform better than the proposed AttnInput for the P@1 setting (up to 9 pinyin length) and the P@5 setting (up to 6 pinyin length).
>
> > What are the potential reasons for the better performance of vanilla RWKV over the proposed approach in the P@1 setting (up to 9 pinyin length) and P@5 setting (up to 6 pinyin length)? How will this affect the practical usage of the proposed model for pinyin input?
>
> we noticed that AttnInput performs slightly worse than vanilla RWKV6 in Top-1 accuracy. This phenomenon is also observed in previous works [1]. Our hypothesis is that the training procedure led to a slight degradation in the original model’s performance. We analyzed instances where the vanilla RWKV6 model provided the correct answer, while AttnInput failed to prioritize the target. Our investigation revealed that in these specific instances, the abbreviated pinyin corresponded to numerous contextually appropriate Chinese character sequences, causing AttnInput to encounter difficulties in accurately ranking them based on probability. This observation supports our initial hypothesis.
>
> ## W4
>
> > Were multiple runs done (with different seeds) for the main results presented in Figure 3? If so, the error bars should be included as well.
>
> No, random seeds do not affect the results because we employ deterministic decoding. (see 3.6 PINYIN-CONSTRAINED TRAINING AND INFERENCE for detailed information)
>
> ## W5
>
> Thank you! All typos have been fixed.
>
> References
>
> [1] [https://aclanthology.org/2022.acl-long.133.pdf](https://aclanthology.org/2022.acl-long.133.pdf)

---

### Official Review · Reviewer_aWjT · 2024-10-27

**Soundness:** 2
**Presentation:** 3
**Contribution:** 2
**Rating:** 5
**Confidence:** 4

**Summary:**

Authors present a novel form of pinyin to Hanzi character conversion model using a RWKV model as the backbone.

**Strengths:**

Language technology support for pinyin is a critical field for investigation but remains niche in NLP community. Conventional wisdom would be to conduct SFT with an LLM, whereas the authors have shown that superior results can be achieved with specialized architectures. This is a suitably novel approach. Further, the use of RWKV instead of a conventional LLM is a necessary consideration of alternative technologies that goes understudied in current environment.

**Weaknesses:**

Several motivations of the paper are no substantiated and would benefit from citation of other works to defend. For instance, authors claim:

"However, inserting pinyin sequences disrupts the semantic flow between the prompt and target text,
poses challenges for effectively leveraging pretrained large language models, as their training objective primarily focuses on predicting the next token."

But lacks substantiation. Especially with current developments in long-context language modeling, there's no reason to suspect an attention based framework to maintain different semantic contexts despite the break in flow.

While investigation of other LLM frameworks is sorely needed in the research community, transformer-decoders are effectively the default when it comes to LLMs. Choice of RWKV in their stead would benefit from clear motivation and comparison against methods.

Use of Top-N for accuracy is an infrequent metric. Taken in combination with the lack of a numbers table and the heavy overlap between models in the Top-1 counts, the performance of the model appears questionable. This issue can be alleviated with a numeric table.

**Questions:**

Could you provide an appendix section on Pinyin? For non-native writers, it's difficult to follow the convention and how it overlaps with Hanzi system.

What was the motivation for using RWKV as opposed to transformer based methods and current SOTA LLM systems?

Could you substantiate the concern for semantic discontinuity in the concatenation approach?

Please provide clear accuracy numbers in support, or in lieu, of the graphs. It is difficult to evaluate if results are significant without.

---

> ### Author Response · Authors · 2024-11-21
>
> We thank you for the feedback and address all remaining concerns below. For further details, please refer to the newly uploaded file, where the modifications are highlighted in blue font. Thank you again for your valuable comments.
> ## Q1
>
> > Could you provide an appendix section on Pinyin? For non-native writers, it's difficult to follow the convention and how it overlaps with Hanzi system.
>
> Thank you for your suggestion! We have already provided the appendix section.
>
> ## Q2
>
> > What was the motivation for using RWKV as opposed to transformer based methods and current SOTA LLM systems?
>
> 1. RWKV uses a unique tokenizer that encodes each Chinese character as a single token, whereas other pre-trained large language models use tokenizers that encode multiple Chinese characters into one token. This has brought us great convenience in training.
> 2. The authors of RWKV6 claim that RWKV6 has ”infinite” context length on https://rwkv.com/ due
>    to the observed continuous decrease in loss as the context length extends beyond the context length used during
>    training. However, this does not necessarily imply that RWKV6 outperforms Transformer-based models in
>    long-text understanding or retrieval tasks.
> 3. RWKV is more efficient during inference compared to transformer based methods.
>
> ## Q3
>
> > Could you substantiate the concern for semantic discontinuity in the concatenation approach?
>
> Sure, we trained a concat-based model with RWKV6-1.6B, labeled as RWKV6-concat-lora. This model was fine-tuned with LoRA  and includes 500M trainable parameters. The training data is the same as the AttnInput model. We tested this model, and its performance was disappointing. (see Figure 3)
>
> The observed inferior performance of RWKV6-concat-lora relative to vanilla RWKV6 provides compelling evidence in support of our proposition that concat-based method disrupts semantic consistency and leads to inefficient training.
>
> ## Q4
>
> > Please provide clear accuracy numbers in support, or in lieu, of the graphs. It is difficult to evaluate if results are significant without.
>
> Thank you for your suggestion! The detailed numeric table is in Appendix C now.

---

> > ### Comment · Reviewer_aWjT · 2024-11-23
> >
> > Q1: This is helpful. But could you provide a bit more detail. At minimum, it should helps the reader understand why in Figure 1. JDGB can map to a single character. Just an example of homophony should suffice.
> >
> > Q2: This is better. Please also add to introduction at least a passing mention of the limitation of vanilla decoder models for thoroughness.
> >
> > Q3: Ah, if you're relying on your empirical results, please clarify that in the introduction would you make this claim. I was confused as if you were responding to previous limitations discovered for the task, not the following results from your work.
> >
> > Q4: Thank you, that's properly thorough.

---

> > > ### Author Response · Authors · 2024-11-23
> > >
> > > Thank you for your detailed response! We have addressed these issues in the latest uploaded PDF. All modifications are highlighted in orange.

---

### Official Review · Reviewer_qBNK · 2024-11-03

**Soundness:** 3
**Presentation:** 3
**Contribution:** 3
**Rating:** 5
**Confidence:** 4

**Summary:**

The paper presents a novel approach to improving the Pinyin Input Method Engine (IME) by leveraging the RWKV language model. AttnInput addresses the integration of Pinyin with large language models, aiming to overcome challenges like semantic discontinuity and inefficient training. The model uses a lightweight side network to enhance the RWKV model's state representations with Pinyin information, improving efficiency in both training and inference. AttnInput claims state-of-the-art performance in abbreviated Pinyin input by utilizing RWKV's linear computational complexity and infinite context length. The proposed method also reduces computational requirements compared to prior approaches like PinyinGPT-Concat. The experimental results showcase significant performance improvements, particularly with longer Pinyin sequences, while maintaining practical latency for real-world applications.

**Strengths:**

The paper proposes an innovative integration of a lightweight side network with the RWKV model to enhance Pinyin IMEs, utilizing RWKV's infinite context length and efficient linear computational complexity, which is novel in the context of Pinyin input. The model achieves state-of-the-art results in experiments, especially with long Pinyin sequences, demonstrating the value of integrating the Pinyin information directly into the language model state. The structure of the paper is well-organized, with detailed explanations of the model components such as RWKV6, AttnInput, ladder side-tuning, and efficient training mechanisms. The experiments are thoroughly explained, with useful visual aids like figures and tables to support the analysis. AttnInput presents a meaningful advancement in the use of large language models for Pinyin IMEs, with potential real-world applications. The reduction in computational resources and training data compared to previous methods, without compromising performance, is a substantial contribution to the field.

**Weaknesses:**

The experimental evaluation is primarily limited to synthetic data generated from SkyPile-150B, which may not fully reflect performance on real-world user-generated text, where varied and noisy input could pose additional challenges. The ablation study only tests the model without Pinyin sequences but does not explore other architectural variations, such as different side network configurations, limiting the insights into each component's importance. Additionally, the paper primarily compares AttnInput with the vanilla RWKV6 and PinyinGPT-Concat, lacking direct comparisons with other state-of-the-art methods like LSTM-based or Transformer-based IMEs, which would provide a broader context for the contribution.

**Questions:**

Can the authors provide more detailed evaluations on real-world datasets, including user-generated Pinyin input, to validate the model's robustness to noisy and diverse input?

How does the performance of AttnInput compare when trained with fewer training steps or using smaller datasets? Does the model generalize well with reduced training resources?

Could the authors expand on the computational cost of integrating ladder side-tuning compared to other parameter-efficient fine-tuning techniques? How does this approach balance trade-offs between performance and efficiency?

---

> ### Author Response · Authors · 2024-11-21
>
> We thank you for the feedback and address all remaining concerns below. For further details, please refer to the newly uploaded file, where the modifications are highlighted in blue font. Thank you again for your valuable comments.
>
> ## Q1
>
> > The experimental evaluation is primarily limited to synthetic data generated from SkyPile-150B, which may not fully reflect performance on real-world user-generated text, where varied and noisy input could pose additional challenges.
>
> > Can the authors provide more detailed evaluations on real-world datasets, including user-generated Pinyin input, to validate the model's robustness to noisy and diverse input?
>
> This paper does not take noise in the input into consideration. We assume that the user's Pinyin input is always correct, which is also the current practice of most input methods.
>
> ## Q2
>
> > The ablation study only tests the model without Pinyin sequences but does not explore other architectural variations, such as different side network configurations, limiting the insights into each component's importance.
>
> > How does the performance of AttnInput compare when trained with fewer training steps or using smaller datasets? Does the model generalize well with reduced training resources?
>
> We compared models trained for 30k steps and 40k steps, and the test results were very similar, indicating that the model generalizes well with reduced training resources.
>
> We did not mention this in the paper because we considered it unimportant. AttnInput requires minimal training resources.
>
> ## Q3
>
> > Could the authors expand on the computational cost of integrating ladder side-tuning compared to other parameter-efficient fine-tuning techniques? How does this approach balance trade-offs between performance and efficiency?
>
> The computational coat of ladder side-tuning is shown in Appendix A. In actuality, the principal merit of ladder side-tuning is its capacity to diminish the memory consumed by activations, consequently facilitating the use of larger batch sizes. Nevertheless, the size of activations depends on multiple factors, including the model's structure and the strategy for recomputation, thus rendering it difficult to analyze with a definitive formula. This is not the focus of this paper, so we did not elaborate on the analysis. In our experiments, ladder side-tuning trained faster than LoRA. The balance between performance and efficiency is achieved by adjusting the number of trainable parameters

---

> ### Author Response · Authors · 2024-11-27
>
> Dear Reviewer,
>
> Thank you for taking the time to review our work and provide your valuable feedback. We welcome any additional comments, questions, or concerns you may have, as it would give us an opportunity to address them further.

---

### Note · Authors · 2025-02-13

I have read and agree with the venue's withdrawal policy on behalf of myself and my co-authors.

---

### Meta-Review · Area_Chair_wz8n · 2024-12-19

**Metareview:**

The paper proposes AttnInput to improve the Pinyin Input Method Engine (IME) by utilizing the RWKV language model. This method addresses key challenges in integrating Pinyin with large language models, such as semantic discontinuity and inefficient training processes.

However, there are noteworthy concerns: 1) the topic of 'Pinyin Input Method Engine' is relatively limited in scope, as evidenced by the scarce literature in Section 5 Relation Work. 2) Several motivations for the 'pinyin input' task have yet to be substantiated (as pointed out by Reviewer aWjT). 3) the motivation for choosing RWKV.

**Additional Comments On Reviewer Discussion:**

1) Conduct more detailed evaluations using real-world datasets. (Reviewer qBNK)
2) Provide direct comparisons with other state-of-the-art methods, such as LSTM-based or Transformer-based IMEs (Reviewer ZXk9, aWjT, qBNK).
3) Several motivations presented in the paper are not substantiated. (Reviewer aWjT)
4) The model's performance appears questionable. (Reviewer aWjT)
5) The presentation is unclear and lacks structure. (Reviewer gCM1)

The authors did not directly address all the questions, and the responses can not dispel my doubts on the above questions.

---

### Decision · Program_Chairs · 2025-01-22

Reject